# Online Inverse Optimal Control for Time-Varying Cost Weights

**DOI:** 10.3390/biomimetics9020084

**Published:** 2024-01-31

**Authors:** Sheng Cao, Zhiwei Luo, Changqin Quan

**Affiliations:** Graduate School of System Informatics, Kobe University, 1-1 Rokkodai-cho, Nada-ku, Kobe 657-8501, Japan; luo@gold.kobe-u.ac.jp (Z.L.); quanchqin@gold.kobe-u.ac.jp (C.Q.)

**Keywords:** inverse optimal control, online calculation, time-varying cost weights, robust to noises

## Abstract

Inverse optimal control is a method for recovering the cost function used in an optimal control problem in expert demonstrations. Most studies on inverse optimal control have focused on building the unknown cost function through the linear combination of given features with unknown cost weights, which are generally considered to be constant. However, in many real-world applications, the cost weights may vary over time. In this study, we propose an adaptive online inverse optimal control approach based on a neural-network approximation to address the challenge of recovering time-varying cost weights. We conduct a well-posedness analysis of the problem and suggest a condition for the adaptive goal, under which the weights of the neural network generated to achieve this adaptive goal are unique to the corresponding inverse optimal control problem. Furthermore, we propose an updating law for the weights of the neural network to ensure the stability of the convergence of the solutions. Finally, simulation results for an example linear system are presented to demonstrate the effectiveness of the proposed strategy. The proposed method is applicable to a wide range of problems requiring real-time inverse optimal control calculations.

## 1. Introduction

The integration of biological principles with robotic technology heralds a new era of innovation, with a significant focus on applying optimal control and optimization methods to analyze animal motion. This approach guides robotic movement development evident in [1], which explores the intricate control systems in mammalian locomotion. Such research underpins the development of robots that emulate the efficiency and adaptability found in nature.

These advancements in understanding animal locomotion through optimal control methods set the stage for the relevance of inverse optimal control (IOC). IOC offers a retrospective analysis of expert movements—human or animal—to infer underlying cost functions optimized in these motions. This methodology is crucial when direct modeling of optimal strategies is complex or unknown.

The use of inverse optimal control (IOC) to identify suitable cost functions from the observable control input and state trajectories of experts is becoming increasingly important. Several successful applications of IOC in estimating the cost weights of multi-features have been reported. For example, the knowledge and expertise of specialists can be categorized and exploited in several fields, including robot control and autonomous driving. The authors of [2], who employed game theory in tailoring robot–human interactions, proposed a method for estimating the human cost function and selecting the robot’s cost function based on the results, leading to the Nash equilibrium in human–robot interactions. The authors of [3] applied IOC to analyze taxi drivers’ route choices. To investigate the cost combination of human motion, the authors of [4] conducted an experiment using IOC techniques to study human motion during the performance of a goal-achieving task using one arm. Additionally, the authors of [5] represented the learning of biological behavior as an inverse linear quadratic regulator (LQR) problem and proposed adaptive methods for modeling and analyzing human reach-to-grasp behavior. Furthermore, the authors of [6] employed an IOC method to segment human movement.

Linear quadratic regulation is a common optimal control method for linear systems. In the 1960s and 1970s, numerous researchers offered solutions to the inverse LQR problem [7,8,9]. Recently, the theory of linear matrix inequality was employed to solve the inverse LQR problem [5,10,11]. Regarding the application of the IOC method for nonlinear systems, several approaches involving methods such as passivity-based condition monitoring [12] or robust design [13] have been reported.

Recent studies in the field of IOC have demonstrated significant advancements. The authors of [14] provided a comprehensive review of the methodologies and applications in inverse optimization, highlighting its growing importance across various domains. The authors of [15] introduced a novel method for sequential calculation in discrete-time systems, enhancing the IOC model’s efficacy under noisy data conditions. The authors of [16] employed a multi-objective IOC approach to explore motor control objectives in human locomotion, which has implications for predictive simulations in rehabilitation technology. Furthermore, the authors of [17] delve into cost uniqueness in quadratic costs and control-affine systems, shedding light on the non-uniqueness cases in IOC. Moreover, a recent thesis [18] introduces a Collage-Based Approach for solving unique inverse optimal control problems, leveraging the Collage method for ODE inverse problems in conjunction with Pontryagin’s Maximum Principle.

Feature-based IOC methods, which involve modeling the cost function as a linear combination of various feature functions with unknown weights, have gained acclaim in recent years [19,20,21,22]. However, it may be difficult to apply these methods to the analysis of complex, long-term behaviors using simple feature functions, e.g., analyzing human jumping [23]. To address this challenge, the authors of [24] proposed a technique for recovering phase-dependent weights that switch at unknown phase-transition points. This method employs a moving window along the observed trajectory to identify the phase-transition points, with the window length determined by a recovery matrix aimed at minimizing the number of observations required for successful cost-weight recovery. Although this method is effective in estimating phase-dependent cost weights, the complex computational requirements limit its use in real-time applications, such as human–robot collaboration tasks. Additionally, in this method, the cost weights in each phase are assumed to be fixed, which may not be generalizable. For example, the human jump motion in [23] was analyzed using time-varying, continuous cost weights.

Overall, the IOC still has several shortcomings that need to be addressed, particularly when applied in approximating complex, multi-phase, continuous cost functions in real time. In this paper, we propose a method for recovering the time-varying cost weights in the IOC problem for linear continuous systems using neural networks. Our approach involves constructing an auxiliary estimation system that closely approximates the behavior of the original system, followed by determining the necessary conditions for tuning the weights of the neurons in the neural network to obtain a unique solution for the IOC problem. We demonstrate that the unique solution corresponds to achieving a zero error between the original system state and the auxiliary estimated system state, as well as zero error between the original costate and the integral of the estimated costate. Based on this analysis, we develop two neural-network frameworks: one for approximating the cost-weight function and the other for addressing the error introduced by the auxiliary estimation system. Additionally, we discuss the necessary requirements for the feature functions to ensure the well-posedness of our online IOC method. Finally, we validate the effectiveness of our method through simulations.

This work makes several significant contributions:We provide a solution for the recovery of time-varying cost weights, essential for analyzing real-world animal or human motion.Our method operates online, suitable for a broad spectrum of real-time calculation problems. This contrasts with previous online IOC methods that mainly focused on constant cost weights for discrete system control.We introduce a neural network and state observer-based framework for online verification and refinement of estimated cost weights. This innovation addresses the critical need for solution uniqueness and robustness against data noise in IOC applications.

## 2. Problem Formulation

### 2.1. System Description and Problem Statement

Consider an object’s system dynamics formulated as
(1)$$\dot{x}=Ax+Bu$$
where $A\in R^{n\times n}$ and $B\in R^{n\times m}$ are two time-invariant matrixes, $x\in R^{n}$ represents the system states, and $u={\left[u_{1},\ldots ,u_{m}\right]}^{T}\in R^{m}$ denotes the control input of the system [25].

To minimize the following cost function while accounting for dynamics (Equation 1), the classic optimal control problem is required to design the optimal control input $u^{*}\left(t\right)$, and generate a sequence of optimal states $x^{*}\left(t\right)$. (Superscript ∗ stands for the optimal condition.)
(2)$$V\left(x,t\right)=\int_{t}^{t_{f}}L_{0}\left(x,u,\tau \right)d\tau$$
Here, $L_{0}$ has the following form:(3)$$L_{0}=q^{T}F\left(x\right)+r^{T}G\left(u\right)$$
where $q={\left[q_{1},q_{2},\ldots ,q_{n_{f}}\right]}^{T}\in R^{n_{f}}$ and $r={\left[r_{1},r_{2},\ldots ,r_{m}\right]}^{T}\in R^{m}\;\forall r_{i}>0$ represent the cost weight vectors, F(x) is referred to as the general union feature vector with respect to *x*, and G(u) indicates the feature vector that is only relevant to the control input *u* [26]. $n_{f}$ represents the feature’s number, which is different from the dimension of system states. For simplicity, we assume that $r^{T}G\left(u\right)=u^{T}Ru$ where *R* is an unknown matrix with $R=\left[\begin{array}{ccc} r_{1} & 0 & \cdots \\ \vdots & \ddots & \vdots \\ \cdots & 0 & r_{m} \end{array}\right]$. Additionally, it is assumed that (A,B) is controllable, *B* is a full column rank matrix, and *A* and *B* are bounded such that $\left||A|\right|\leq \delta_{A}\;\;||B||\leq \delta_{B}$.

### 2.2. Maximum Principle in Forward Optimal Control

To minimize the cost function as is the case in (Equation 2) with $L_{0}$ defined in (Equation 3), there exists a costate variable vector λ that satisfies Pontryagin’s maximum principle as follows:(4)$$\begin{array}{c} \dot{\lambda }=-{\bar{F}}_{x}^{T}q-A^{T}\lambda \end{array}$$(5)$$\begin{array}{c} Ru+B^{T}\lambda =0 \end{array}$$
where ${\bar{F}}_{x}=\frac{\partial F(x)}{\partial x}$ and $\lambda \in R^{n}$ denote the costate variables. These two equations are derived from Pontryagin’s Maximum Principle by taking the partial derivatives of the Hamiltonian function defined by $H\left(x,u,\lambda \right)=L_{0}+\lambda^{T}\left(Ax+Bu\right)$, specifically $\dot{\lambda }=-\frac{\partial H}{\partial x}$ and $\frac{\partial H}{\partial u}=0$. The initial value of λ can be represented as $\lambda_{0}$.

The optimal control input $u^{*}$ of the system expressed by (Equation 1) is given as
(6)$$u^{*}=-R^{-1}B^{T}\lambda$$
where λ is unknown. Thus, using this optimal control input, we have
(7)$$\dot{x}=Ax-H\lambda$$
where *H* denotes the matrix $H=BR^{-1}B^{T}$. Notably, given that *B* is a full column rank matrix, it is clear that *H* is invertible. In addition, since *B* is a bounded constant matrix, there exists a positive scalar $\delta_{H}$ such that *H* satisfies $\left||H|\right|\leq \delta_{H}$.

Additionally, the time derivatives of the system dynamics can be formulated as follows:(8)$$\ddot{x}=A\dot{x}-H\dot{\lambda }$$

### 2.3. Analysis of the IOC Problem

We assume that the system states $x[t,t_{f}]$ and the control input $u[t,t_{f}]$, which represent the time series of the system states and control inputs from time point *t* to $t_{f}$, provide the solution to the optimal minimization of the cost function (Equation 2). In addition, we assume that the optimal system states and control input satisfy the boundary conditions $\left||x|\right|\leq \delta_{x}$
$\left||u|\right|\leq \delta_{u}$
$\left|\right|\dot{u}\left|\right|\leq \delta_{\dot{u}}$.

The objective of the IOC problem is to recover the unknown cost weight’s vector q(t). Furthermore, IOC, for example, may be employed to analyze different behaviors such as the effect of different occasions on the relative importance of certain human motion feature functions. A rigorous analysis of the derived cost weights that can recreate the original data $x\left[t,t_{f}\right],u\left[t,t_{f}\right]$ is required for the aforementioned applications. To begin, we consider two problems:What happens when a different feature function is selected?

In previous studies, it was assumed that the cost weight vector *q* is either a constant value [19] or a step function with multiple phases [24]. These assumptions have been effective in recovering the cost weights used in the analysis of optimal control methods for a robot’s motion control, such as analyzing the motion of a robot controlled by a LQR approach. However, occasionally, it may be inappropriate to assume that the cost weights are constants or step functions when analyzing the complex behaviors of natural objects, such as human motion. In particular, deciding which feature function to adopt when evaluating the motion of natural objects could pose a challenge.

**Proposition** **1.**
*Depending on the different selections of feature functions F(x) for the IOC, the original constant cost weight q may become a time-varying continuous function.*


**Proof.** From (Equation 8), for the objects’ original feature function, we have
(9)$$\begin{array}{c} H^{-1}\left(-\ddot{x}+A\dot{x}+HA^{T}H^{-1}Bu\right)={\bar{F}}_{ox}^{T}q_{o} \end{array}$$
where $q_{o}$ denotes the original time-invariant cost weight vector, and ${\bar{F}}_{o}\left(x\right)$ denotes the partial derivative with respect to *x* of the original feature function. When we choose a different feature function $F_{n}\left(x\right)$, the above equation becomes
(10)$$\begin{array}{c} H^{-1}\left(-\ddot{x}+A\dot{x}+HA^{T}H^{-1}Bu\right)={\bar{F}}_{nx}^{T}q_{n} \end{array}$$
where ${\bar{F}}_{nx}$ denotes the partial derivative with respect to *x* of the new selected feature function and $q_{n}$ is the corresponding cost weights on ${\bar{F}}_{nx}$. Thus, we have
$${\bar{F}}_{ox}^{T}q_{o}={\bar{F}}_{nx}^{T}q_{n}\;\forall t_{0}\leq t\leq t_{f}$$From this equation, it follows that $q_{n}$ may be a time-varying function when ${\bar{F}}_{ox}$ and ${\bar{F}}_{nx}$ are not equivalent, and as ${\bar{F}}_{ox}$ and ${\bar{F}}_{nx}$ are continuous functions, we can reasonably conclude that $q_{n}$ is also a continuous function.    □

Based on this proposition, it is crucial to expand the definition of cost weights to include time-varying values, as this will facilitate a more accurate analysis of the motion of increasingly complex natural objects. Despite the need for time-varying cost weight recovery in many applications, it has received minimal research attention thus far.
Whether or not the given set $x\left[t,t_{f}\right],u\left[t,t_{f}\right]$ in the IOC problem has a unique solution {q(t),r}.
The uniqueness of the solution to the IOC problem when cost weights are constant has been discussed in many studies [15,17,18,22]. In this work, we determine if there is still a unique solution to the IOC problem when *q* is a time-varying function.

From (Equation 10), we can find different continuous functions q(t) such that the equation is satisfied for different values of *R* (different values of *H*). This implies that if *q* is considered as a time-varying function, the set {q(t),r} will not have a unique solution.

Therefore, when we consider the unique solution of the IOC problem with the time-varying function q(t), it is necessary to introduce additional conditions to ensure that the IOC problem has a unique solution and that the resulting unique solution is meaningful.

In this study, for simplicity, we assume that R=I [27,28], where *I* is the identity matrix. In actual optimal control cost functions, when we focus on reducing one of the control inputs $u_{i}$, the convergence of the i-th system state $x_{i}$ related to $u_{i}$ will also be affected. Consequently, the final control result shows that the change in each state of the system is not solely influenced by the chosen cost weights q(t), but also by R(t). In the IOC problem, setting R(t)=I allows the effect of different weights on different control inputs in the original system to be reflected in the current estimate of q(t). This enables us to view the estimated weights on the system states as representing the relative importance of each state in the system’s dynamic evolution, without considering the impact of the control input on these weights.

Based on our conclusion that *q* may be time-varying when different feature functions are chosen and on the corresponding conditions under which a unique solution exists, we can define the IOC problem to be solved in this study as follows:

**Problem** **1.**
*Online Estimation of Time-Varying Cost weights q(t)*




*
**Given**
*

*: (1) Measured system state x as well as control input u (2) R=I*




*
**Goal**
*

*: Online estimate of the time-varying q(t) utilizing the given x and u.*


## 3. Adaptive Observer-Based Neural Network Approximation of Time-Varying Cost Weights

In this study, we estimate time-varying cost weight functions online using an observer-based adaptive neural network estimation approach, as opposed to earlier studies that required a large number of time series of *x* and *u* to recover fixed cost weights offline.

### Construction of the Observer

Following the introduction of $\hat{q}\left(t\right)\in R^{n}$ denoting the estimation of q(t), we define the estimation of the associated costate variable $\hat{\lambda }$ as follows:(11)$$\dot{\hat{\lambda }}=-{\bar{F}}_{\hat{x}}^{T}\hat{q}\left(t\right)-A^{T}\hat{\lambda }$$
where ${\bar{F}}_{\hat{x}}=\frac{\partial F(\hat{x})}{\partial \hat{x}}$ denotes the partial derivatives of the feature functions that are only relevant to the estimated system states $\hat{x}$ obtained by inserting $\hat{\lambda }$ into (Equation 7):(12)$$\dot{\hat{x}}=A\hat{x}-H\hat{\lambda }$$
where the initial state ${\hat{x}}_{0}$ of this system is selected to be ${\hat{x}}_{0}=x_{0}$.

Thus, compared with that of the original system, the error generated by the new estimation system can be expressed as
(13)$$\dot{\tilde{x}}=A\tilde{x}-H\tilde{\lambda }$$
(14)$$\dot{\tilde{\lambda }}=-{\bar{F}}_{x}^{T}q\left(t\right)+{\bar{F}}_{\hat{x}}^{T}\hat{q}\left(t\right)-A^{T}\tilde{\lambda }$$
where $\tilde{\lambda }=\lambda -\hat{\lambda }$ and $\tilde{x}=x-\hat{x}$. Here, the feature function is selected such that its partial derivative with respect to *x* is bounded and it is assumed that $\left|\right|{\bar{F}}_{x}^{x}\left|\right|\leq \delta_{nx}$, $\left|\right|{\bar{F}}_{\hat{x}}\left|\right|\leq \delta_{n\hat{x}}$ and $\left|\right|{\bar{F}}_{x}^{x}\left(x\right)-{\bar{F}}_{\hat{x}}\left(\hat{x}\right)\left|\right|\leq \zeta ||\tilde{x}\left|\right|$ where $\delta_{nx}$, $\delta_{n\hat{x}}$ and ζ denote a positive scalar.

Additionally, the time derivatives of (Equation 14) can be expressed as
(15)$$\begin{array}{c} \ddot{\tilde{x}}=A\dot{\tilde{x}}-H\dot{\tilde{\lambda }} \end{array}$$

Thus, the following equation can be satisfied:(16)$$\begin{array}{c} \dot{s}=A_{r}s+T_{x}\tilde{q}+\left(T_{x}-T_{\hat{x}}\right)\hat{q} \end{array}$$
where $s=\left[\begin{array}{c} \dot{\tilde{x}}\\ \tilde{\lambda } \end{array}\right]$, $A_{r}=\left[\begin{array}{cc} A & HA^{T}\\ 0 & -A^{T} \end{array}\right]$, $T_{x}=\left[\begin{array}{c} H{\bar{F}}_{x}\\ -{\bar{F}}_{x} \end{array}\right]$, $T_{\hat{x}}=\left[\begin{array}{c} H{\bar{F}}_{\hat{x}}\\ -{\bar{F}}_{\hat{x}} \end{array}\right]$. $\tilde{q}$ denotes the error of estimating *q*. Here, $\left|\right|{\bar{F}}_{x}^{x}\left(x\right)-{\bar{F}}_{\hat{x}}\left(\hat{x}\right)\left|\right|\leq \zeta ||\tilde{x}\left|\right|$ implies that there exists a positive scalar $\zeta^{\prime }$ such that $\left|\right|T_{x}-T_{\hat{x}}\left|\right|\leq \zeta^{\prime }\left|\right|\tilde{x}\left|\right|$ holds. Based on the bound of ${\bar{F}}_{x}^{x}\left(x\right),{\bar{F}}_{\hat{x}}\left(\hat{x}\right),H$, it follows that there are two positive scalars $\delta_{t_{x}}$ and $\delta_{t_{\hat{x}}}$ such that the following inequalities hold: $\left|\right|T_{x}\left|\right|\leq \delta_{t_{x}}$ and $\left|\right|T_{\hat{x}}\left|\right|\leq \delta_{t_{\hat{x}}}$.

Moreover, from (Equation 6) and (Equation 7), λ can be calculated as follows:(17)$$\lambda =-H^{-1}Bu$$

## 4. Neural Network-Based Approximation of Time Varying Cost Weights

In this section, a neural network-based cost weight approximation algorithm is proposed. To calculate an approximation of the time-varying vector *q*, we adopt a neural network in which the chosen inputs are $u_{I}=\left[\begin{array}{c} x_{0}\\ u \end{array}\right]$, where $x_{0}$ denotes the initial state of the system (Equation 1). Based on this, we assume that time-invariant weight matrixes $W\in R^{n_{f}\times l}$ exist that satisfy the following expression:(18)$$\begin{array}{c} q=W^{T}\phi \left(u_{I}\right)+\epsilon_{1}\left(u_{I}\right) \end{array}$$
where $\phi (u_{I})$ denotes the activation function and $\epsilon_{1}\left(u_{I}\right)$ denotes the structure approximation error of the neural networks. In addition, the activation function selected enables the activation function as well as its partial derivative to satisfy the following boundary condition: $||\phi \left(u_{I}\right)\left|\right|\leq \delta_{p}$ and $\left|\right|\frac{\partial \phi (u_{I})}{\partial u_{I}}\left|\right|\leq \delta_{pu}$ where $\delta_{p}$ and $\delta_{pu}$ represent two positive scalars. Additionally, $\left|\right|\epsilon_{1}\left(u_{I}\right)\left|\right|\leq \epsilon_{n}$ where $\epsilon_{n}$ is a positive scalar.

The estimate of vector *q* is constructed as follows:(19)$$\begin{array}{c} \hat{q}={\hat{W}}^{T}\phi \left(u_{I}\right) \end{array}$$
where $\hat{W}$ denotes the estimation of *W*. In this paper, we will combine two estimators ${\hat{W}}_{1}$ and ${\hat{W}}_{2}$ to estimate *W*, as shown in Section 4.1. Before presenting the details of the estimators, we first discuss the necessary conditions for the estimation.

Based on the setting of estimator $\hat{W}$, the error of estimating *q* can be expressed as
(20)$$\begin{array}{c} \tilde{q}=q-\hat{q}={\tilde{W}}^{T}\phi \left(u_{I}\right)+\epsilon_{1}\left(u_{I}\right) \end{array}$$
where $\tilde{W}=W-\hat{W}$ denotes the error of estimating *W*. Substituting $\tilde{q}$ into (Equation 16) yields
(21)$$\begin{array}{c} \dot{s}=A_{r}s+T_{x}{\tilde{W}}^{T}\phi \left(u_{I}\right)+\left(T_{x}-T_{\hat{x}}\right)\hat{q}+T_{x}\epsilon_{1}\left(u_{I}\right) \end{array}$$

To profoundly comprehend the necessary condition for the convergence of the estimation error $\tilde{W}$, we define uniformly ultimately bounded (UUB) below.

**Definition** **1.**
*A time-varying signal σ(t) can be said as UUB if there exists a compact set $S\subset R^{n}$ so that for all σ∈S, there exists a bound μ≥0 and a time T such that ||σ||≤μ for all $t\geq t_{0}+T$.*


**Lemma** **1.**
*If the following conditions are satisfied, $\tilde{W}$ becomes UUB.*

*$\int_{t_{0}}^{t_{i}}sdt,s$ become UUB after a time point $t_{1}$ ($\left|\right|\int_{t_{0}}^{t_{i}}sdt\left|\right|\leq \delta_{1}$, and $\left||s|\right|\leq \delta_{2}$)*

*The change in $\hat{W}$ approaches zero*

*Matrix C defined below will become a full row rank matrix.*


(22)
$$\begin{array}{c} C=\left[\begin{array}{c} \int_{t_{1}+1}^{t_{1}+2}T_{x}{\left(I⊗\phi \left(u_{I}\right)\right)}^{T}dt\\ \vdots \\ \int_{t_{i}-1}^{t_{i}}T_{x}{\left(I⊗\phi \left(u_{I}\right)\right)}^{T}dt \end{array}\right] \end{array}$$

*where $t_{1}\leq t_{i}\leq t_{f}$ and any term in C satisfies the persistent excitation (PE) condition defined below.*

(23)
$$\begin{array}{c} \left|\right|\int_{t_{j}}^{t_{j}+1}T_{x}\left(I⊗\phi \left(u_{I}\right)\right){dt)}^{T}\left|\right|\geq \beta_{j}\;\;\forall t_{1}\leq t_{j}\leq t_{i} \end{array}$$

*Here, $\beta_{j}$ is a positive value.*


**Proof.** From (Equation 21)
(24)$$\begin{array}{cc} s & =A_{r}\int_{t_{0}}^{t_{i}}sdt+\int_{t_{0}}^{t_{i}}T_{x}{\tilde{W}}^{T}\phi \left(u_{I}\right)dt\\ \; & +\int_{t_{0}}^{t_{i}}\left(T_{x}-T_{\hat{x}}\right)\hat{q}dt+\int_{t_{0}}^{t_{i}}T_{x}\epsilon_{1}\left(u_{I}\right)dt \end{array}$$Since $\int_{t_{0}}^{t_{i}}sdt\to 0,s\to 0$ reaches a steady state and $A_{r}$ is a constant, we can obtain the following:
(25)$$\begin{array}{c} ||s-A_{r}\int_{t_{0}}^{t_{i}}sdt\left|\right|\leq \delta_{si} \end{array}$$
where $\delta_{si}$ denotes a small positive scalar. Additionally, with both $\epsilon_{1}\left(u_{I}\right)$ and $T_{x}$ being bounded, this leads to
(26)$$\begin{array}{c} \left|\right|\int_{t_{0}}^{t_{i}}T_{x}\epsilon_{1}\left(u_{I}\right)dt\left|\right|\leq \delta_{T\epsilon } \end{array}$$
where $\delta_{T\epsilon }$ denotes a small positive scalar. The term $\int_{t_{0}}^{t_{i}}T_{x}\epsilon_{1}\left(u_{I}\right)dt$ captures the effect of the structural error of the neural network on state *s*. Since $T_{x}$ is bounded, when the neural network approximates the cost weight function adequately, the value of $\epsilon_{1}\left(u_{I}\right)$ decreases, which in turn minimizes the overall integral value. In other words, a well-selected neural network structure with a good approximation of the cost weight function will produce a small structure error and, therefore, a small overall integral value $\int_{t_{0}}^{t_{i}}T_{x}\epsilon_{1}\left(u_{I}\right)dt$.(Equation 24)–(Equation 26) leads to
(27)$$\begin{array}{c} \left|\right|\int_{t_{0}}^{t_{i}}T_{x}{\tilde{W}}^{T}\phi \left(u_{I}\right)dt+\int_{t_{0}}^{t_{i}}\left(T_{x}-T_{\hat{x}}\right)\hat{q}dt\left|\right|\leq \delta_{si}+\delta_{T\epsilon } \end{array}$$Similarly, we can obtain a similar relation for the duration $\left[t_{0},t_{1}\right]$
(28)$$\begin{array}{c} \left|\right|\int_{t_{0}}^{t_{1}}T_{x}{\tilde{W}}^{T}\phi \left(u_{I}\right)dt+\int_{t_{0}}^{t_{1}}\left(T_{x}-T_{\hat{x}}\right)\hat{q}dt\left|\right|\leq \delta_{si}+\delta_{T\epsilon } \end{array}$$From (Equation 27) and (Equation 28), it follows that   
(29)$$\begin{array}{c} \left|\right|\int_{t_{1}+1}^{t_{i}}T_{x}{\tilde{W}}^{T}\phi \left(u_{I}\right)dt+\int_{t_{1}+1}^{t_{i}}\left(T_{x}-T_{\hat{x}}\right)\hat{q}dt\left|\right|\leq 2\left(\delta_{si}+\delta_{T\epsilon }\right) \end{array}$$Furthermore, considering $\int_{t_{0}}^{t_{i}}sdt\to 0$ after $t_{1}$, the definition of *s* and $\left|\right|T_{x}-T_{\hat{x}}\left|\right|\leq \zeta^{\prime }\left|\right|\tilde{x}\left|\right|$, this implies that
(30)$$\begin{array}{cc} \left|\right|\int_{t_{1}+1}^{t_{i}}\left(T_{x}-T_{\hat{x}}\right)\hat{q}dt\left|\right| & \leq \int_{t_{1}+1}^{t_{i}}\left|\right|\left(T_{x}-T_{\hat{x}}\right)\left|||\right|\hat{q}\left|\right|dt\\ & \leq \int_{t_{1}+1}^{t_{i}}\zeta^{\prime }\delta_{\tilde{x}}\delta_{\hat{q}}dt\equiv \delta_{\zeta (t_{i}-t_{1}-1)} \end{array}$$
where $\delta_{\tilde{x}}$ and $\delta_{\hat{q}}$ represent the bounds of $\tilde{x}$ and $\hat{q}$ respectively. Thus, this leads to the inequality
(31)$$\begin{array}{c} \left|\right|\int_{t_{1}+1}^{t_{i}}T_{x}{\tilde{W}}^{T}\phi \left(u_{I}\right)dt\left|\right|\leq 2\left(\delta_{si}+\delta_{T\epsilon }\right)+\delta_{\zeta (t_{i}-t_{1}-1)} \end{array}$$In this case, when $\dot{\hat{W}}$ approaches zero, the following relation emerges:
(32)$$\begin{array}{cc} \; & \left|\right|\int_{t_{1}+1}^{t_{i}}T_{x}{\left(I⊗\phi \left(u_{I}\right)\right)}^{T}vec\left(\tilde{W}\right)dt\left|\right|\\ \; & =\left|\right|\int_{t_{1}+1}^{t_{i}}T_{x}{\left(I⊗\phi \left(u_{I}\right)\right)}^{T}dt\;vec\left(\tilde{W}\right)\left|\right|\\ \; & \leq 2\left(\delta_{si}+\delta_{T\epsilon }\right)+\delta_{\zeta (t_{i}-t_{1}-1)} \end{array}$$
Based on this relation, it follows that
(33)$$\begin{array}{c} \left|\right|\int_{t_{1}+1}^{t_{1}+2}T_{x}{\left(I⊗\phi \left(u_{I}\right)\right)}^{T}dt\;vec\left(\tilde{W}\right)\left|\right|\\ \leq 2\left(\delta_{si}+\delta_{T\epsilon }\right)+\delta_{\zeta (1)} \end{array}$$
where $\delta_{\zeta (1)}=\int_{t_{1}+1}^{t_{1}+2}\zeta^{\prime }\delta_{\tilde{x}}\delta_{\hat{q}}dt=\cdots =\int_{t_{i}-1}^{t_{i}}\zeta^{\prime }\delta_{\tilde{x}}\delta_{\hat{q}}dt$.Thus, it implies that
(34)$$\begin{array}{c} ||C\;vec\left(\tilde{W}\right)\left|\right|\leq \left(t_{i}-t_{1}-1\right)\left(2\left(\delta_{si}+\delta_{T\epsilon }\right)+\delta_{\zeta (1)}\right) \end{array}$$
where *C* is defined in (Equation 22). Due to *C* being full row rank, this leads to
(35)$$\begin{array}{cc} \left||vec(\tilde{W})|\right| & \leq \left|\right|C^{+}||||C\;vec\left(\tilde{W}\right)\left|\right|\\ \; & \leq \left|\right|C^{+}\left|\right|\left(t_{i}-t_{1}-1\right)\left(2\left(\delta_{si}+\delta_{T\epsilon }\right)+\delta_{\zeta (1)}\right) \end{array}$$
From (Equation 23), we have $\left|\right|C^{+}\left|\right|\leq \frac{1}{\sqrt{\left(t_{i}-t_{1}-1\right)\beta_{j}^{2}}}$
(36)$$\begin{array}{c} ||vec\left(\tilde{W}\right)\left|\right|\leq \sqrt{\frac{t_{i}-t_{1}-1}{\beta_{j}^{2}}}\left(2\left(\delta_{si}+\delta_{T\epsilon }\right)+\delta_{\zeta (1)}\right) \end{array}$$
Thus, $\tilde{W}$ is UUB.Notably, $\beta_{j}$ evaluates the lower bound of the norm of $\int_{t_{j}}^{t_{j}+1}T_{x}\left(I⊗\phi \left(u_{I}\right)\right){dt)}^{T}$, it can increase when the data *x* cause the norm of the integral to deviate significantly from zero. The size of $\delta_{\zeta (1)},\delta_{si}$ is related to the minimization of *s* and $\int_{t_{0}}^{t_{i}}sdt$, and the size of $\delta_{T\epsilon }$ is related to the approximation ability of the chosen neural network. The bound of $\tilde{W}$ after $t_{1}$ can be minimized by the excited *x*, successfully minimizing *s* and $\int_{t_{0}}^{t_{i}}sdt$ while appropriately designing the structure of the neural network.    □

### 4.1. Construction of the Neural Network

As shown in Lemma 1, the convergence of $\int_{t_{0}}^{t}sd\tau$ is essential in the convergence of $\tilde{W}$ to 0. Therefore, it is necessary to incorporate this consideration in the approximation design.

First, we divide the estimation of the weights of the neural network into two parts:(37)$$\hat{W}={\hat{W}}_{1}+{\hat{W}}_{2}$$
and
(38)$$\begin{array}{c} \hat{q}={\hat{q}}_{1}+{\hat{q}}_{2}={\left({\hat{W}}_{1}+{\hat{W}}_{2}\right)}^{T}\phi \left(u_{I}\right) \end{array}$$
where ${\hat{q}}_{1}={\hat{W}}_{1}^{T}\phi \left(u_{I}\right)$ and ${\hat{q}}_{2}={\hat{W}}_{2}^{T}\phi \left(u_{I}\right)$.

The necessity for employing two distinct estimators, ${\hat{W}}_{1}$ and ${\hat{W}}_{2}$, is rooted in their specialized roles in minimizing the tracking error *s*. This dual-estimator approach ensures that $\hat{q}\left(t\right)$ closely aligns with the desired trajectory q(t). While ${\hat{W}}_{1}$’s adaptive tuning is primarily aimed at steering *s* towards zero, its inherent residual errors in its adaptive process necessitate the deployment of ${\hat{W}}_{2}$ for error compensation and enhanced accuracy in tracking the ideal cost weight q(t). To gain a deeper understanding of this system, we will begin by examining the error dynamics, which forms a fundamental basis for the subsequent detailed exploration of the tuning laws for each estimator.

The state equation describing the error dynamics can be obtained as follows:(39)$$\dot{s}=A_{r}s+T_{x}{\tilde{q}}_{1}+\left(T_{x}-T_{\hat{x}}\right){\hat{q}}_{1}-T_{\hat{x}}{\hat{q}}_{2}$$
where $s=\left[\begin{array}{c} \dot{\tilde{x}}\\ \tilde{\lambda } \end{array}\right]$, $A_{r}=\left[\begin{array}{cc} A & HA^{T}\\ 0 & -A^{T} \end{array}\right]$, $T_{x}=\left[\begin{array}{c} H{\bar{F}}_{x}\\ -{\bar{F}}_{x} \end{array}\right]$, $T_{\hat{x}}=\left[\begin{array}{c} H{\bar{F}}_{\hat{x}}\\ -{\bar{F}}_{\hat{x}} \end{array}\right]$.

Further, to effectively minimize $\int_{t_{0}}^{t}sd\tau$, we define vector *e* as follows:(40)$$e=\left(T_{x}-T_{\hat{x}}\right){\hat{q}}_{1}+Ks+K_{p}\int_{t_{0}}^{t}sd\tau -T_{\hat{x}}{\hat{q}}_{2}+A_{r}s$$
where $K=diag\left(\left[k,\cdots ,k\right]\right)\in R^{2n\times 2n}$ and $K_{p}=diag\left(\left[k_{p},\cdots ,k_{p}\right]\right)\in R^{2n\times 2n}$. Parameters *k* and $k_{p}$ are two positive scalars, thus, (Equation 39) can be written as:(41)$$\dot{s}=-Ks-K_{p}\int_{t_{0}}^{t}sd\tau +T_{x}{\tilde{q}}_{1}+e$$

We suppose that an ideal time-invariant weight matrix $W_{2}\in R^{n_{f}\times l}$ exists, which guarantees that
(42)$$\begin{array}{cc} \; & \left(T_{x}-T_{\hat{x}}\right){\hat{q}}_{1}+Ks+A_{r}s+K_{p}\int_{t_{0}}^{t}sd\tau \\ \; & =T_{\hat{x}}q^{\prime }=T_{\hat{x}}\left(W_{2}^{T}\phi \left(u_{I}\right)+\epsilon_{2}\left(u_{I}\right)\right) \end{array}$$
where $u_{I}=\left[\begin{array}{c} x_{0}\\ u \end{array}\right]$.

The estimation error of the neural network can be represented as
(43)$$\begin{array}{cc} {\tilde{q}}_{1} & \equiv q-{\hat{q}}_{1}={\tilde{W}}_{1}^{T}\phi \left(u_{I}\right)+\epsilon_{1}\left(u_{I}\right)\\ {\tilde{q}}_{2} & \equiv q^{\prime }-{\hat{q}}_{2}={\tilde{W}}_{2}^{T}\phi \left(u_{I}\right)+\epsilon_{2}\left(u_{I}\right) \end{array}$$
and *e* can be represented as
(44)$$e=T_{\hat{x}}\left({\tilde{W}}_{2}^{T}\phi \left(u_{I}\right)+\epsilon_{2}\left(u_{I}\right)\right)$$

Therefore, (Equation 41) becomes
(45)$$\begin{array}{cc} \dot{s} & =-Ks-K_{p}\int_{t_{0}}^{t}sd\tau \\ \; & +T_{x}\left({\tilde{W}}_{1}^{T}\phi \left(u_{I}\right)+\epsilon_{1}\left(u_{I}\right)\right)+T_{\hat{x}}\left({\tilde{W}}_{2}^{T}\phi \left(u_{I}\right)+\epsilon_{2}\left(u_{I}\right)\right) \end{array}$$

### 4.2. Tuning Law of the Neural Network for the Estimation of q(t)


An updating law for a neural network that estimates q(t) can be represented in Theorem 1, based on the error system’s dynamics that were derived in (Equation 45).

**Theorem** **1.**
*If we choose the updating laws for the neural network weights ${\hat{W}}_{1}$ and ${\hat{W}}_{2}$ as shown in (Equation 46), respectively, where $\Gamma_{1}$, $\Gamma_{2}$, and $k_{e}$ are positive scalar constants, then state s, $\int_{t_{0}}^{t}sd\tau$ and error e will be UUB.*

(46)
$$\begin{array}{cc} {\dot{\hat{W}}}_{1} & =\Gamma_{1}\phi \left(u_{I}\right)s^{T}T_{x}\\ {\dot{\hat{W}}}_{2} & =\Gamma_{2}\phi \left(u_{I}\right){\left(s+k_{e}e\right)}^{T}T_{\hat{x}} \end{array}$$

*In addition, if there exist positive constants $t_{\delta }$, $\beta_{1}$, $\beta_{2}$, $\beta_{3}$, and $\beta_{4}$ such that the inequalities in (Equation 47) are satisfied for all initial times $t_{0}$, then the signals ${\tilde{W}}_{1}$ and ${\tilde{W}}_{2}$ will also be UUB.*

(47)
$$\begin{array}{cc} \beta_{2}I & \geq \int_{t_{0}}^{t_{0}+t_{\delta }}C_{p1}{\left(t\right)}^{T}C_{p1}\left(t\right)dt\geq \beta_{1}I\\ \beta_{4}I & \geq \int_{t_{0}}^{t_{0}+t_{\delta }}C_{p2}{\left(t\right)}^{T}C_{p2}\left(t\right)dt\geq \beta_{3}I \end{array}$$

*Here, $C_{p1}\left(t\right)=T_{x}\left(I⊗\phi {\left(u_{I}\right)}^{T}\right)$, $C_{p2}\left(t\right)=T_{\hat{x}}\left(I⊗\phi {\left(u_{I}\right)}^{T}\right)$*


**Proof.** A proof of this theorem can be found in Appendix A.    □

Applying (Equation 46) results in *s*, $\int_{t_{0}}^{t}sd\tau$, and *e* being UUB, as shown in Theorem 1. Additionally, (Equation 46) shows that when *s* and *e* decreases, ${\dot{\hat{W}}}_{1}$ and ${\dot{\hat{W}}}_{2}$ decrease as well, resulting in a decrease in $\dot{\hat{W}}={\dot{\hat{W}}}_{1}+{\dot{\hat{W}}}_{2}$. At this point, as stated in Lemma 1, if the condition of matrix *C* (defined in Lemma 1), being a full row rank matrix, is satisfied, then $\tilde{W}={\tilde{W}}_{1}+{\tilde{W}}_{2}$ will also be UUB. Thus, the solution to the IOC problem can be derived by applying (Equation 38).

## 5. Simulations

### 5.1. Basic Simulation Conditions

To verify the effectiveness of our method, we performed the simulations using a sample linear system controlled by the optimal control method with the original cost weights *R* selected in two cases.

The sample linear system dynamics can be formulated as follows:(48)$$\dot{\theta }=A\theta +B\tau$$
where $\theta ={\left[\theta_{1},\theta_{2}\right]}^{T}\in R^{2}$ represents the system states. We select $A=\left[\begin{array}{cc} 30 & 80\\ 60 & 0 \end{array}\right]$, $B=\left[\begin{array}{cc} 2 & 0\\ 0 & 4 \end{array}\right]$ and $\tau \in R^{2}$ denoting the control input.

The cost function selected in these simulations is formulated as
(49)$$\begin{array}{c} V_{r}=\frac{1}{2}\int_{0}^{t_{f}}\left(\theta^{T}Q\left(t\right)\theta +\tau^{T}R\tau \right)dt \end{array}$$
when all the elements of θ satisfying $|\theta_{i}|\leq \theta_{r_{l}}$ and $Q\left(t\right)=\left[\begin{array}{cc} q_{1} & 0\\ 0 & q_{2} \end{array}\right]$ is the continuous time-varying cost weights on system states θ. $R=\left[\begin{array}{cc} r_{1} & 0\\ 0 & r_{2} \end{array}\right]$ represents the cost weights on the control inputs.

Moreover, in our simulations, we select 0 as the initial value of all the elements of both ${\hat{W}}_{1}$ and ${\hat{W}}_{2}$. Actuation function $\phi (u_{I})$ was selected as $\phi \left(u_{I}\right)={\left[\phi_{1}\left(u_{I}\right),\cdots ,\phi_{i}\left(u_{I}\right),\cdots ,\phi_{l}\left(u_{I}\right)\right]}^{T}$ with $\phi_{i}\left(u_{I}\right)$ designed as
(50)$$\phi_{i}\left(u_{I}\right)=exp\left(\frac{-{\left(u_{I}-\psi_{i}\right)}^{T}\left(u_{I}-\psi_{i}\right)}{\nu }\right)$$
where ν denotes a positive scalar and $\psi_{i}$ denotes the center of the respective activation function. We initialized the activation function centers on a four-dimensional grid to match the dimension of $u_{i}$, ensuring a uniform distribution across the input space and enhancing network adaptability.

The overall implementation for recovering the time-varying cost weights is shown in Algorithm 1.
**Algorithm 1** Online implementation**Input****:** 
$\left\{x_{i},u_{i}\right\}$**Output****:** 
$\hat{q}\left(t\right)$*Initialization*:1:Initialize $\hat{\lambda }$, $\hat{x}$, ${\hat{W}}_{1}$, ${\hat{W}}_{1}$, ${\hat{W}}_{2}$, $\Gamma_{1}$, $\Gamma_{2}$ and R=I.*LOOP Process*2:**for** i=0 to *K* **do**3:   Calculate λ using $\lambda =-H^{-1}Bu$.4:   Calculate $\dot{\hat{x}}$ and $\dot{\hat{\lambda }}$ using (Equation 11) and (Equation 12).5:   Calculate $\dot{\tilde{x}}$ and $\dot{\tilde{\lambda }}$ using (Equation 14) and (Equation 13).6:   Calculate $s=\left[\begin{array}{c} \dot{\tilde{x}}\tilde{\lambda } \end{array}\right]$.7:   Calculate *e* following (Equation 40).8:   Calculate $\phi (u_{I})$ and update ${\hat{W}}_{1}$, ${\hat{W}}_{2}$ using (Equation 46).9:   Calculate $\hat{q}\left(t\right)$ using (Equation 38).10:**end for**11:**return**$\;\hat{q}\left(t\right)$

Two cases are considered in the simulation:In the first case, we apply the optimal control of the sample system with cost weights θ as the signal ($q_{1}\left(t\right)=1+cos\left(t\right)$ and $q_{2}\left(t\right)=2+sin\left(t\right)$). The proposed IOC method is employed online to estimate the cost weights, with the simultaneous online recovery of the original system trajectory. Parameters $\Gamma_{1}$ and $\Gamma_{2}$ in the updating law are set to $\Gamma_{1}=1$ and $\Gamma_{2}=1$, respectively. Parameters *k* and $k_{p}$ are set to k=50 and $k_{p}=625$, respectively. The initial values of ${\hat{W}}_{1}$ and ${\hat{W}}_{2}$ are set to matrixes with all elements equal to zero. The original $r_{1}$ and $r_{2}$ are set to $r_{1}=1$ and $r_{2}=1$, respectively. The simulation also uses 49 nodes in the neural network.In the second case, we perform the simulation of our IOC method, but with the original $r_{1}$ and $r_{2}$ set to $r_{1}=3$ and $r_{2}=4$, respectively. All other simulation settings are the same as in the first case.

Similar to the simulation sections in previous works ([6,24]), we use the control input from the simulation, which ignores the measurement issues with the control input and measurement errors that may occur in real-world applications. This allows us to purely evaluate the performance of our method in solving the IOC problem. In actual applications, the control input can be calculated by substituting the measured $\dot{\theta }$ into (Equation 48), as described in [24].

### 5.2. Results

The simulation results are shown in the figures below.

In Figure 1, the blue solid line represents the original variation in the cost weights whereas the gray solid line represents the estimated cost weights. After a brief period of oscillation at the initial time, our method accurately recovers the original cost weights when R=I. Notably, similar to the case in other adaptive control methods and adaptive neural network based control methods, the initial oscillation is a result of the adaptive initialization of the weights in (Equation 46) due to the large initial errors in ${\tilde{W}}_{1}$ and ${\tilde{W}}_{2}$.

Figure 2 demonstrates the impact of selecting R=I on the estimation results when the original *R* value is arbitrary. The solid blue line represents the original time-varying cost weights, whereas the dotted gray line represents the final estimated values. Although the estimated values differ from the original values, the general trend of the changes is preserved. In addition, the gray line represents the mutual weights in the dynamics of the system state, whereas the original weights among the control inputs are reflected in the current estimate of q(t). From the figure, we can observe that the bottom lines in blue and gray colors represent the value of the original and estimated $q_{2}$. Evidently, the blue line for $q_{2}$ is larger than that for $q_{1}$ from 4.8 s to 5 s. Additionally, in the original settings, $r_{2}$ is 4, which confers greater importance to the decrease in $u_{2}$ compared with the case when $r_{1}=3$, leading to the weakening of the convergence of the $\theta_{2}$ term associated with $u_{2}$. In our estimates, the value of the dashed line for the estimated $q_{2}$, which also considers the impact from original setting of *R* is not greater than the value of estimated $q_{1}$ between 4.8 s and 5 s. This indicates that the convergence of $\theta_{2}$ is weakened by considering the impact from the cost weights on control input. Our dashed line more accurately reflects the actual situation compared to the blue line.

In Figure 3, Figure 4 and Figure 5, we show the results of error *e*, states *s* and $\int_{t_{0}}^{t}sd\tau$ in two cases. The blue lines show the results of the first case, whereas the gray dotted lines show the results of the second case. From the figures, we can observe that all the values effectively decrease to a low range during the simulation, and most importantly, in the second case, the different selections of *R* do not affect the convergence of these values. This demonstrates the effectiveness of our method and highlights that even with different values of *R*, the recovered cost weights are still feasible solutions to the IOC problem, as they can be utilized to regenerate a similar system trajectory and control inputs ($\int_{t_{0}}^{t}sd\tau =\left[\begin{array}{c} \tilde{x}\\ \int_{t_{0}}^{t}\tilde{\lambda }d\tau \end{array}\right]\to 0$).

## 6. Discussion

### 6.1. Robustness of the Proposed Method to Noisy Data

In (Equation 46), $\Gamma_{1}$ and $\Gamma_{2}$ decrease the error by regulating the updating speed of the estimated values. Adjusting these two terms may successfully reduce the impact of data noise to a certain degree. Their roles are similar to that of a low-pass filter’s time constant. For example, in the setting of the first case, when noise exists, $x∼N(0,10^{-1})$ and $u∼N(0,10^{-4})$, the simulation results show that different sets of $\Gamma_{1}$ and $\Gamma_{2}$ (e.g., $\Gamma_{1}=10$, $\Gamma_{2}=10$; $\Gamma_{1}=1$, $\Gamma_{2}=1$) can significantly influence the noise reduction performance.

As shown in Figure 6, while relatively small values of $\Gamma_{1}$ and $\Gamma_{2}$ may result in a low convergence rate, they effectively reduce the impact of data noise. Our method demonstrates robustness against noise by allowing for the adjustment of parameters $\Gamma_{1}$ and $\Gamma_{2}$.

### 6.2. Calculation Complexity and Real-Time Calculation

The proposed algorithm has a low computational complexity, as it only involves the calculation of dot products between matrixes and vectors as well as the summation of vectors. Additionally, it does not require any iterative or optimization calculations. This makes it an efficient solution for real-time calculations. In fact, our simulation shows that a single iteration of the algorithm using case 1 settings takes only approximately 0.23 ms in Matlab 2016b to complete the SIOC’s calculation, which is fast enough to meet real-time calculation requirements.

### 6.3. Advantages of Using R=I

The simulation results suggest that one of the key advantages of setting *R* as a constant *I* is that it effectively consolidates the impact of cost weights on state convergence, which would have been influenced by different settings of *R*, into the estimated value of q(t). This allows for a comprehensive evaluation of the system state convergence, as it only depends on q(t), without needing to account for additional considerations. Furthermore, by maintaining a consistent value of R=I, it is possible to standardize the analysis of the same motion across multiple agents, which is crucial for various applications.

## 7. Conclusions

In this paper, we proposed a neural network based method for recovering the time-varying cost weights in the IOC problem for linear continuous systems. Our approach involved constructing an auxiliary estimation system that closely approximates the behavior of the original system, followed by determining the necessary conditions for tuning the weights of the neurons in the neural network to obtain a unique solution for the IOC problem. We discussed the necessary requirements for the previous settings to ensure the well-posedness of our online IOC method. We showed that the unique solution corresponds to achieving a nearly zero error between the original system state and the auxiliary estimated system state, as well as nearly zero error between the original costate and the integral of the estimated costate. Based on this analysis, we developed two neural network frameworks: one for approximating the cost weight function and the other for addressing the error introduced by the auxiliary estimation system and terms. Finally, we validated the effectiveness of our method through simulations, highlighting its ability to recover time-varying cost weights and its robustness against different original choices of *R*. Overall, our method represents a significant advancement in the field of online IOC, and it is applicable to a wide range of problems requiring real-time IOC calculations.

## Figures and Tables

**Figure 1 biomimetics-09-00084-f001:**
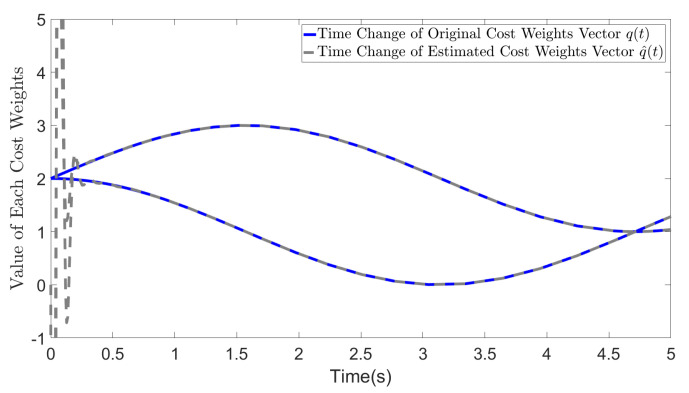
Estimated cost weights ($r_{1}=1,\;r_{2}=1$).

**Figure 2 biomimetics-09-00084-f002:**
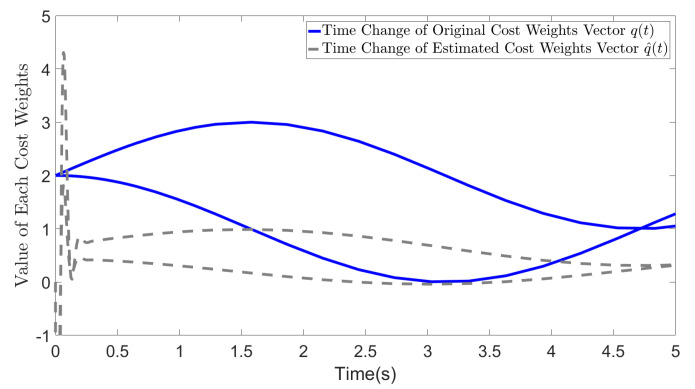
Estimated cost weights ($r_{1}=3,\;r_{2}=4$).

**Figure 3 biomimetics-09-00084-f003:**
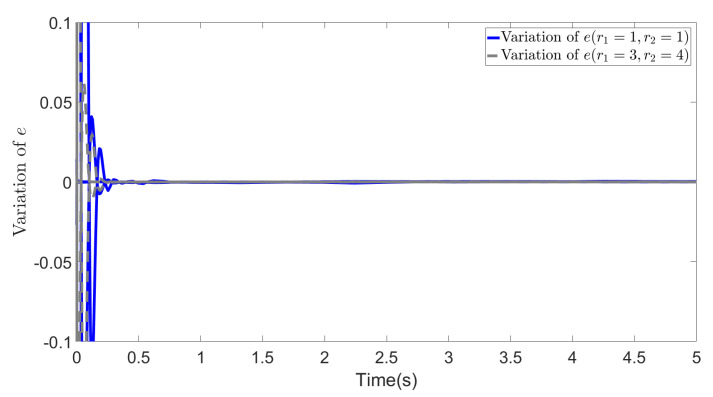
Variation of error *e* ($r_{1}=1,\;r_{2}=1$ and $r_{1}=3,\;r_{2}=4$).

**Figure 4 biomimetics-09-00084-f004:**
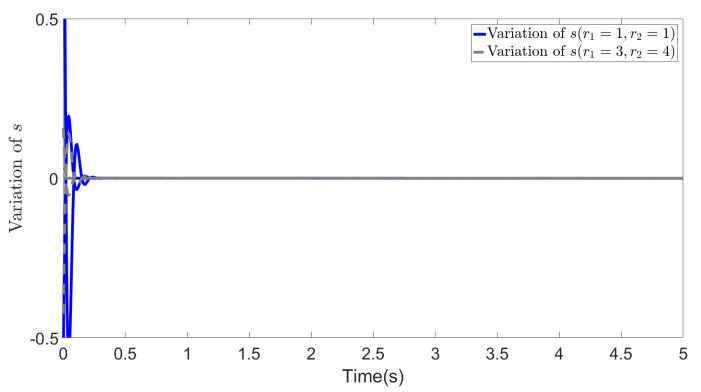
Variation of error *s* ($r_{1}=1,\;r_{2}=1$ and $r_{1}=3,\;r_{2}=4$).

**Figure 5 biomimetics-09-00084-f005:**
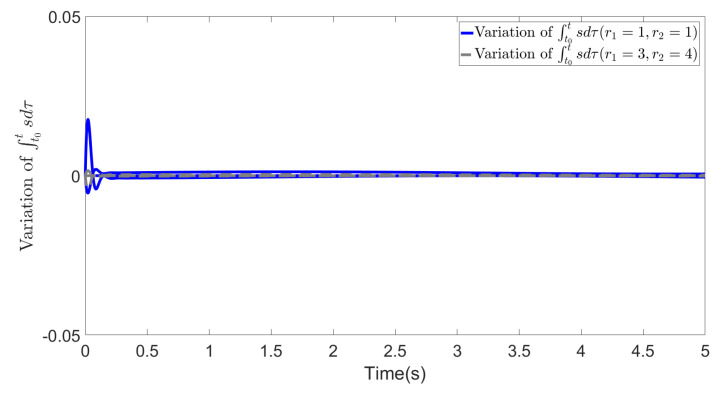
Variation of $\int_{t_{0}}^{t}sd\tau$ ($r_{1}=1,\;r_{2}=1$ and $r_{1}=3,\;r_{2}=4$).

**Figure 6 biomimetics-09-00084-f006:**
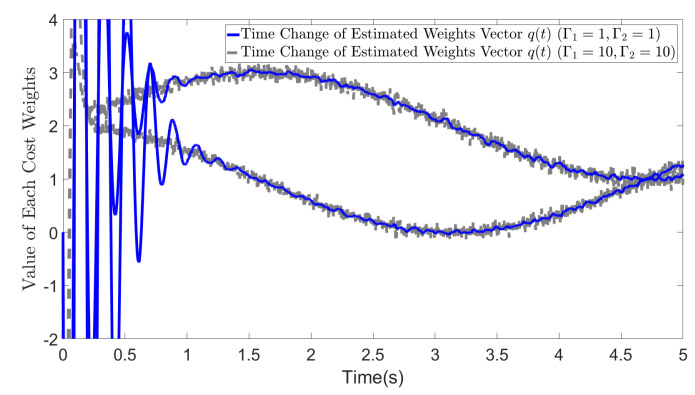
Estimated cost weights (Noisy Case): (1) $\Gamma_{1}=10$, $\Gamma_{2}=10$ (2) $\Gamma_{1}=1$, $\Gamma_{2}=1$.

## Data Availability

Data are contained within the article.

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
