# Peer review of "Online Inverse Optimal Control for Time-Varying Cost Weights"

_biomimetics, 2024, doi:10.3390/biomimetics9020084_

Round 1

Reviewer 1 Report

Comments and Suggestions for Authors

 Comments and Suggestions are indicated in the upload file.

Comments on the Quality of English Language

 Minor editing of English language required

Reviewer 2 Report

Comments and Suggestions for Authors

Dear authors, 

Authors propose an adaptive online approach for inverse optimal control, employing neural-network approximation for time-varying cost weights. 

Typos:

Page 2, line 88: please add one reference to equation (1);

Page 3, line 100: add a reference to justify r^t G(u) and specify why R is unknown;

Page 4, line 151: authors claim that -..” uniqueness of the solution to the IOC…” please add a reference about the uniqueness solution;

Page 6, line 210 and 212: please clarify the notation and write carefully the meaning for all readers: estimating W with two estimators and notations;

Page 8, equation (36): explain why authors need to divide the estimation of the weights.

Page 9, Line 267: what authors suppose that ideal time-invariant? And if it is not considered that, what happens?

Round 2

Reviewer 1 Report

Comments and Suggestions for Authors

Thank you for your answers

Comments on the Quality of English Language

 Minor editing of English language required